# Quantifying the effects of antibiotic resistance and within-host competition on strain fitness in *Streptococcus pneumoniae*

**Aswin Krishna**[1], **Gerry Tonkin-Hill**[2,3,4], **Thibaut Morel-Journel**[5], **Stephen Bentley**[6], **Paul Turner**[7], **François Blanquart**[8], **Sonja Lehtinen**[1,9,10]*

**1** Institute of Integrative Biology, Department of Environmental Systems Science, ETH Zurich, Zurich, Switzerland, **2** Peter MacCallum Cancer Centre, Melbourne, Victoria, Australia, **3** Sir Peter MacCallum Department of Oncology, The University of Melbourne, Melbourne, Victoria, Australia, **4** Department of Microbiology and Immunology, The University of Melbourne, at the Peter Doherty Institute for Infection and Immunity, Melbourne, Victoria, Australia, **5** Université Sorbonne Paris Nord, Université Paris Cité, INSERM, IAME, Bobigny, France, **6** Wellcome Sanger Institute, Cambridge, United Kingdom, **7** Centre for Tropical Medicine and Global Health, Nuffield Department of Medicine, University of Oxford, Oxford, United Kingdom, **8** Center for Interdisciplinary Research in Biology, CNRS, Collège de France, PSL Research University, Paris, France, **9** Department of Computational Biology, University of Lausanne, Lausanne, Switzerland, **10** Swiss Institute of Bioinformatics, Lausanne, Switzerland

* sonja.lehtinen@unil.ch

**Data availability statement:** The data on pneumococcal carriage in infants was obtained

## Abstract

Competition plays a key role in shaping the structure and diversity of bacterial populations. In many clinically important bacterial species, strains compete at multiple scales: at the between-host scale for new hosts to colonise, and at the within-host scale during co-colonisation. Characterising these multiple facets of competition plays an important role in understanding bacterial ecology. This is particularly relevant for antibiotic resistance, where competition between antibiotic-susceptible and resistant strains determines resistance dynamics. In this work, we perform survival analyses on a large longitudinal dataset of *Streptococcus pneumoniae* carriage to quantify how within-host competition affects the rates of clearance and establishment of pneumococcal strains. We find that the presence of a within-host competitor is associated with a 33% increase in clearance and a 54% reduction in establishment. Priority effects and serotype differences partially predict the outcomes of this within-host competition. Further, we quantify the effects of antibiotic resistance on between- and within-host components of fitness. Antibiotic consumption is associated with increased clearance rate for both susceptible and resistant strains, albeit to a higher extent in susceptible strains. In the absence of antibiotics, we find some evidence that resistance is associated with increased susceptibility to within-host competition, suggesting a fitness cost of resistance. Overall, our work provides quantitative insights into pneumococcal competition across scales and the role of this competition in shaping pneumococcal epidemiology.

from https://github.com/gtonkinhill/pneumo_withinhost_manuscript/tree/main/data/epi. The code corresponding to results is available here: https://doi.org/10.5281/zenodo.15800560.

**Funding:** The study was funded by a Swiss National Science Foundation (SNSF) grant to SL. The grant number is PR00P3_201618. Funder website: https://www.snf.ch/en SNSF has not played any role in study design, data analysis or publication decision with respect to this manuscript.

**Competing interests:** The authors have declared that no competing interests exist.

## Introduction

Several clinically important bacteria such as *Escherichia coli* and *Streptococcus pneumoniae* exhibit remarkable strain-level diversity. Strains of these species differ in key traits like virulence, mobility, metabolism or antibiotic-resistance [1]. Interactions between strains affect strain dynamics and thus the frequency of the traits that they carry. If strains are ecologically equivalent (a 'neutral' model), strain dynamics are driven purely by random fluctuations [2,3]. However, in species like *S. pneumoniae* and *E. coli*, both experimental and observational evidence suggests the presence of competition between strains [4–7]. Characterising and quantifying these competitive interactions *in natura* is therefore important for understanding strain dynamics and diversity.

Strains of a species interact across multiple scales. While competition is conceived and modelled in many studies in terms of competition for acquisition of uncolonised hosts [8–12], strains can also compete for growth and survival within a host [13]. In species like *E. coli* and *S. pneumoniae* where co-colonisation is common (e.g. median number of co-colonisers is around 1.5 in pneumococcal carriers [14] and 3 in *E. coli* carriers [15]), within-host competition potentially plays a major role in shaping strain dynamics. However, the specific effects of within-host competition on strain fitness at the epidemiological scale–e.g. its impact on duration of carriage, transmission to other hosts or the efficiency of colonisation–are not well understood. Previous modelling work has demonstrated that assumptions about these effects have significant impacts on population dynamics [16–23], highlighting the importance of this knowledge gap.

In this work, we study competition by quantifying its effects on the ability of strains to establish and persist in hosts (Fig 1). We focus on *S. pneumoniae*, a mostly asymptomatic coloniser of the nasopharynx that can become pathogenic when it evades host immune

| How does this ecological factor... | ... affect these components of fitness? | | |
|---|---|---|---|
| | affects → Clearance rate | Establishment rate | Within-host competitive ability |
| Presence of competitors | *Increases* | *Decreases* | *Not applicable* |
| Cost of antibiotic resistance | *Confounded by association of resistance with low clearance rate* | *Not testable in this data* | *Decreases for some antibiotics* |

**Fig 1. In this study, we are interested in how the presence of within-host competitors and cost of antibiotic resistance shapes the different components of fitness of *S. pneumoniae* strains.** The table here summarises the evidence we find for associations between each ecological factor (in columns) and the fitness measures (in rows).

responses [24]. *S. pneumoniae* is associated with a substantial disease burden in children and geriatric patients [25,26] and features among the World Health Organization (WHO) list of priority pathogens for antibiotic resistance [27]. Our focus is in particular on two aspects of pneumococcal competition: competition between serotypes (i.e. antigenic strain types defined based on the bacteria's polysaccharide capsule) and competition between antibiotic susceptible and resistant strains. Serotype competition is important because it determines how pneumococcal populations respond to vaccinations covering a subset of serotypes. The removal of vaccine-targetted serotypes drives the increase in prevalence of non-targetted serotypes, with major consequences for the impact of the vaccine on e.g. disease prevalence [5,28,29]. Competition between antibiotic-susceptible and resistant strains is important because it determines the behaviour of resistance frequencies [18,30,31].

In the case of antibiotic resistance, the ecology of strain competition is complex. The success of a resistant strain in a host depends on the fitness cost of resistance in the absence of antibiotics and the fitness benefit under antibiotic pressure. These effects can impact different components of strain fitness. At the epidemiological scale, fitness costs could manifest as reduced transmission rate, decreased duration of carriage, and in co-colonised hosts, as a decreased ability to compete with other strains. Numerous studies have estimated the fitness cost of resistance in experimental settings (e.g. [32]), typically in terms of effects on the growth rate. However, it is unclear how these estimates translate to humans *in vivo*, or how a reduction in growth rate affects epidemiological components of strain fitness. While fitness costs at the epidemiological scale can be inferred by fitting transmission models to surveillance and genomic data [33–36], these estimates depend on specific model structures. Untangling the effects of antibiotic resistance on various aspects of strain fitness in absence and presence of antibiotics remains a challenge.

To address these questions, we use a rich longitudinal dataset of carriage of *S. pneumoniae* in infants [14,37,38] to quantify the impact of within-host competition on various aspects of pneumococcal dynamics. The dataset has presence-absence information on pneumococcal serotypes carried in 737 infants sampled monthly for up to 2 years, along with additional information on antibiotic consumption in the infants and resistance phenotypes to seven antibiotics for a subset of all samples. We use survival analyses to quantify the effects of the presence of within-host competitors on two important epidemiological processes - clearance rate and establishment rate of serotypes, thereby linking the two scales of competition. Using the data on resistance status of strains, we quantify the within-host fitness cost and benefit of resistance. Our results advance our understanding of within-host competition between bacterial strains and provide a basis for improved prediction of pneumococcal—and in particular antibiotic resistance—dynamics.

## Results

### Survival models can explain pneumococcal carriage dynamics

We use data on pneumococcal carriage in 737 infants from [14], with nasopharyngeal swabs taken roughly once a month to identify the serotypes that they carried. Assuming complete accuracy of serotype detection, we infer 5529 episodes of colonisation by 68 different serotypes across all infants (see Fig A in S1 Supplementary information for a visualization of the data). We use these inferred colonisation episodes to quantify the effects of within-host processes on two epidemiological parameters in our analyses: clearance rate and establishment rate. Clearance rate is defined as the inverse of the average time between establishment of a strain and its clearance (i.e. the inverse of the duration of carriage). The establishment rate of a strain is defined as the inverse of the average time to establishment of the strain since the

previous clearance of that strain in the host (Fig 2). This rate is related to, but not the same as, transmission rate: establishment rate is the product of the transmission rate and the density of transmitting hosts.

We use Cox Proportional Hazards modelling to quantify the effects of different strain-associated factors on clearance and establishment rates in independent models. Briefly, the Cox model describes the time to an event T (time to clearance, or time to establishment) using a hazard function as follows:

$$H(t|X) = H_0(t) \exp(\beta_1 X_1 + \beta_2 X_2 + ... + \beta_p X_p) \tag{1}$$

Here, $H$ denotes the hazard of clearance or establishment, $H_0$ denotes the baseline hazard function, $X_p$ denotes the value of covariate $p$, and $\beta_p$ represents the strength of covariate $p$ in determining the hazard of the event. The function $H_0$ is not estimated by the Cox model.

**Diversity in clearance rates across serotypes.** First, we introduce age of infant at colonisation and serotype of the colonising pneumococci in a model for time to clearance. The different serotypes are included as covariates in the model and age is included as a stratifying variable (here, and in all subsequent analyses, see Methods). Stratification by age leads to an age-dependent baseline hazard function in the model. The hazard coefficients corresponding to the serotypes describe the inherent durations of carriage of the different serotypes. Serotypes with a lower hazard coefficient are those with a longer duration of carriage—i.e. a lower clearance rate. The results of this analysis (summarised in Fig Ba and Bb in S1 Supplementary information) capture known patterns of pneumococcal ecology. For example, serotypes 6B, 19F and 23F, with the lowest hazard coefficients in the analysis, are also known to be long-colonisers from previous studies [39,40].

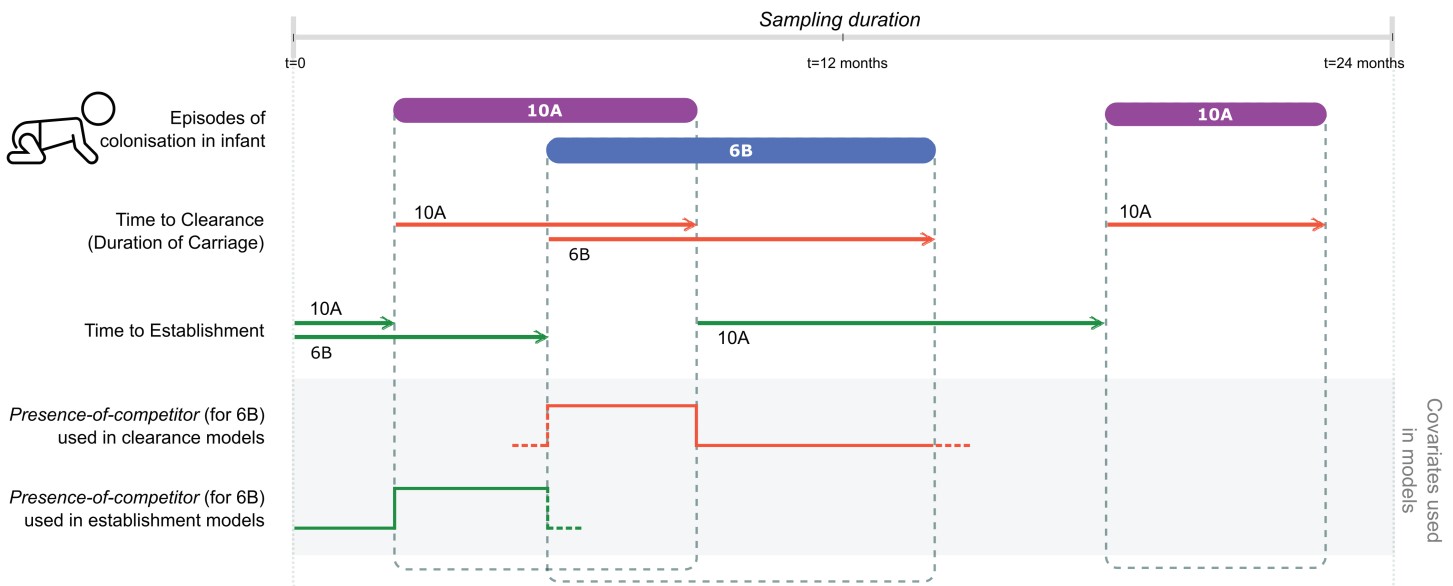

**Fig 2. Definitions of covariates used in the survival analyses.** Monthly sampling data of infants is used to identify the presence of serotypes over time in hosts, and infer the time to clearance and establishment of serotypes. Time to clearance is the duration from establishment to clearance of a serotype. Time to establishment is the duration from last clearance of that serotype to establishment. *Presence-of-competitor* variable is used to indicate whether there exists a competing strain in the host either during the time to clearance (orange lines) or time to establishment (green lines).

**Antibiotic consumption increases clearance rates.** Next, we introduce information on antibiotic use in infants into the above model (see Methods for details about the data). We expect to observe faster clearance of serotypes in hosts due to antibiotic consumption. We define a time-dependent binary variable *drug-use* that tracks whether an infant consumes any antibiotic during each episode of colonisation. Here, the start dates of *drug-use* are taken from our dataset and the length of all treatments are assumed to be 7 days based on WHO guidelines on treatment of children [41] (sensitivity to the assumption discussed in Note G in S1 Supplementary information). We include *drug-use* as a time-varying covariate, serotypes as constant covariates, and age as a stratifying variable in the model. To enable model convergence, we removed serotypes with zero variance in survival outcomes (these are serotypes with low prevalence) from the analysis (see Methods). The results (Fig Bc in S1 Supplementary information) reveal a positive hazard coefficient of 0.48 (95% confidence intervals: CI = 0.37 to 0.59) corresponding to *drug-use*. The exponential of the hazard coefficient yields a hazard ratio of 1.62 (95% CI: 1.45-1.80), indicating that antibiotic consumption increases the hazard of strain clearance by 62%.

## Within-host competition increases clearance rate of *S. pneumoniae*

We then look at the effect of within-host competition on the clearance rate of a serotype. To do this, we define a time-dependent variable *presence-of-competitor*, which tracks the presence of one or more co-colonising serotypes in addition to the focal serotype. *Presence-of-competitor* corresponding to a focal serotype at a given time equals 1 if there exists another serotype in the same host at that time (Fig 2). This is a binary variable: we assume that the effect on the focal serotype is independent of the number of co-colonising serotypes or their abundances. We perform time-varying Cox proportional hazards regression using *presence-of-competitor* as the time-varying covariate along with serotype of the focal strain, while correcting for infant age by stratification (Fig 3A). We use these covariates since the model that includes serotype has a lower partial-AIC compared to one that does not (details in Note I in S1 Supplementary information). The hazard coefficient corresponding to *presence-of-competitor* is 0.28 (CI = 0.21 to 0.35), indicating an increase in clearance rate in the presence of another serotype (Fig 2A)—here, a hazard coefficient of zero would indicate that a co-colonising serotype does not impact clearance rate, corresponding to a 'neutral' model. Thus, co-colonisation is associated with an increased risk of clearance of exp(0.28) = 1.33 (CI = 1.24 to 1.42), or 33%.

Next, we looked further into what determines the outcomes of within-host competition. To test whether serotypes differ in their within-host competitive abilities, we introduce interaction terms between each serotype and *presence-of-competitor*. These interaction terms capture the extent to which the clearance rate of each focal serotype is affected by the presence of another competing serotype. We perform a time-varying Cox proportional hazards model with the following covariates: *presence-of-competitor*, serotype of the focal strain, and all of the interaction terms. To determine if the intrinsic clearance rate (i.e. clearance rate in the absence of competitors) of a serotype predicts the effect of a co-colonising competitor, we fit a linear model (using orthogonal distance regression that accounts for uncertainty in both variables—see Methods) between the hazard coefficients of serotypes and their interaction terms. We find a negative relationship (slope=-0.40, CI = -0.60 to -0.20) between inherent rate of clearance and the change in clearance rate under within-host competition (Fig 3B). These results suggest that faster clearing strains may be better at within-host competition than longer-carried ones, indicating a potential trade-off between intrinsic duration of carriage and

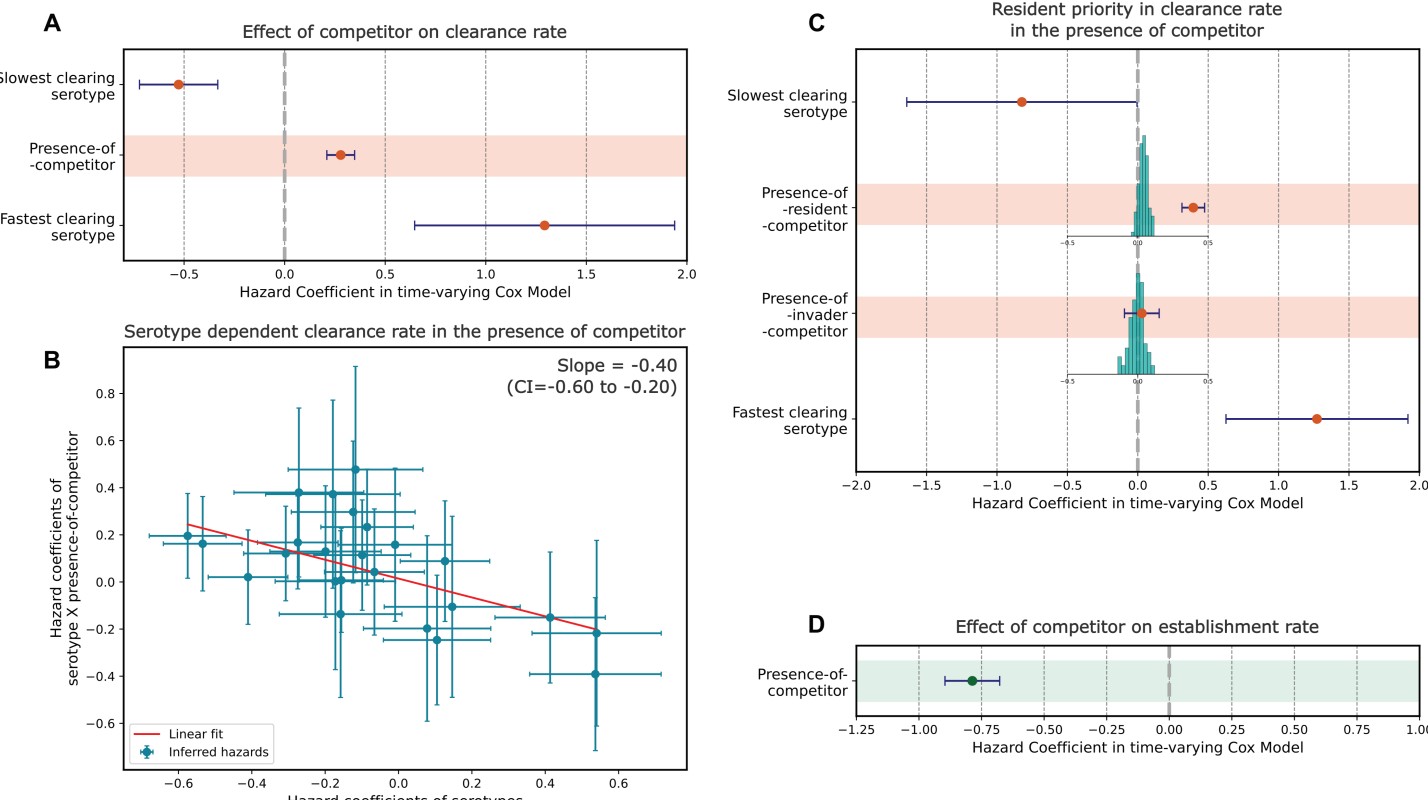

**Fig 3. Models that show the effects of within-host competition.** A–C correspond to models for clearance rate, and D corresponds to a model for establishment rate. Error bars show 95% confidence intervals. (A) Does the presence of a competitor affect the clearance rate of a serotype? *Presence-of-competitor* has a positive hazard, indicating increased clearance rate in the presence of competitors. (B) Does the increase in clearance in the presence of competitors depend on the serotype of the focal strain? The plot shows a linear model that maps hazard coefficients of serotypes to hazard coefficients of interaction terms between serotype and *presence-of-competitor*. Each blue dot corresponds to one serotype. Error bars in this plot are standard errors. The red line is the fit to a linear model. (C) Do resident strains have a competitive advantage for clearance? *Presence-of-resident-competitor* is associated with an increased clearance rate, while *presence-of-invader-competitor* is not. The histograms represent the null distribution of the variables when episodes of colonisation occur randomly within infants. (D) Does the presence of a competitor affect establishment rate of a strain? *Presence-of-competitor* has a negative hazard on establishment, indicating a reduced rate of establishment into an occupied host niche. The data underlying this figure can be found in https://doi.org/10.5281/zenodo.15800560

within-host competitive ability. Overall, the above analysis indicates that serotype-specific competitive abilities can partially predict the outcome of within-host competition.

We also tested whether order of colonisation of strains in the host can affect clearance under competition. We refer to this as a priority effect, in which a resident strain can have a competitive advantage in clearance over a newly established strain. We replace *presence-of-competitor* with two covariates that encompass information on order of establishment of the serotypes: *presence-of-resident-competitor* and *presence-of-invader-competitor*. *Presence-of-invader-competitor* indicates that the focal strain is the resident serotype and conversely, *presence-of-resident-competitor* indicates the focal serotype is the invader. A Cox proportional hazards model with serotype of the focal strain, *presence-of-resident-competitor* and *presence-of-invader-competitor* as covariates reveals that the presence of a resident serotype increases the risk of clearance of the competitor significantly more than an invader serotype (Fig 3C). This analysis is potentially confounded because acquisition of an additional serotype is more likely to occur during longer episodes of carriage—long episodes of carriage are therefore

more likely to be resident than short ones. To correct for this effect, we compare the coefficients from our analysis to their expected null distributions when episodes of colonisation in infants are randomly reshuffled (histograms in Fig 3C). This reshuffling allows us to generate the null distributions of the hazard coefficients in absence of competitive effects, i.e. where effects are only due to the confounding described above. The coefficient of *presence-of-resident-competitor* (0.40, CI = 0.32 to 0.48) is significantly higher than that expected from its null distribution (p-value<0.01). The presence of an invader strain is not associated with an increase in clearance rate. This analysis suggests that pneumococcal within-host competition is asymmetric, with the invader strain incurring a higher clearance risk compared to the resident strain. This result is qualitatively sensitive to how the start and end of a colonisation episode are defined and to the assumption that within-host diversity is fully sampled (see Notes B and C in S1 Supplementary information).

## Within-host competition reduces establishment rate of *S. pneumoniae*

Next, we look at the effect of within-host competition on the rate of establishment of a serotype. The establishment rate of a serotype is defined in terms of the time since the last clearance event of that serotype from the host (Fig 2). We test whether the establishment rate is modified by the presence of a resident competitor in the host. Here, we are interested in the presence of potential competitors in the host during the 'time to establishment' of the serotype. We define a time-dependent variable named *presence-of-competitor* that varies along the time to establishment of a serotype, and tracks the status of the host into which it eventually establishes: in the time interval preceding establishment of the focal serotype, *presence-of-competitor* equals one when the host is carrying at least one other strain (i.e. non-focal serotype). As in the previous analyses, we run a time-varying Cox proportional hazards model on the data using *presence-of-competitor* as a time-varying covariate. We do not include the serotype of the incoming strain as a covariate in this analysis since serotype-differences in establishment rates depend on the prevalence of the serotypes in the population. Instead, we add serotype along with age of infant as a stratification variable that allows for a different baseline hazard function for each serotype. This adjusts for the effect of serotype without estimating its effect on the outcome.

This analysis (Fig 3D) reveals a significant decrease in establishment rate in the presence of additional serotypes in the host. The *presence-of-competitor* covariate has a negative hazard coefficient of -0.79 (CI =-0.90 to -0.68) and a hazard ratio of 0.46 (CI = 0.41 to 0.51). This corresponds to establishment into a host occupied by a competitor occurring at about half the rate (reduction by 1-0.46 = 54%, CI = 49% to 59%) compared to establishment into an empty host niche.

Although the effect of serotype of the incoming focal strain could not be estimated, we can test whether the serotype of the resident competitor affects the establishment rate of the focal strain. We run a time-varying Cox proportional hazards model using *presence-of-competitor* and serotype of the resident strain as covariates, while adjusting for age of host and serotype of the focal strain by including them as stratification variables. The model shows variability in how well resident serotypes prevent colonisation by a new serotype (Fig E in S1 Supplementary information). Serotypes with a negative hazard coefficient may be better at competition during colonisation and preventing establishment of incoming serotypes, while those with a positive coefficient may be worse at preventing establishment. A likelihood ratio test shows an overall significant effect of serotype of competitor on establishment rate (*p-value:* $1.2 \times 10^{-13}$). In summary, the serotype of within-host competitors can affect the rate of establishment of incoming pneumococcal strains.

## Antibiotics and antibiotic resistance affect clearance rates of strains

Next, we shift our focus to understanding how antibiotic consumption and antibiotic resistance affect pneumococcal dynamics. Antibiotic consumption is associated with an overall 62% increase in clearance rate *(Section: Survival models can explain pneumococcal carriage dynamics)*, but this effect may depend on the resistance profiles of these strains. Conversely, the effect of resistance on clearance and establishment can depend on antibiotic consumption: resistance is potentially beneficial under antibiotic exposure, but costly in the absence of antibiotics. We seek to separately quantify the effects of drug consumption and resistance on clearance rates in our dataset (effects on establishment discussed in Note D in S1 Supplementary information). In this section, we focus on the effects of antibiotic consumption on the dynamics of antibiotic-resistant versus susceptible strains.

The dataset contains information on 13 different drug treatments that the infants are exposed to, but resistance status of the strains to only seven antibiotics. We map the drug treatments to the corresponding resistance phenotypes based on antibiotic class (see Methods), and define a categorical variable for drug use that takes into account both the resistance status of the focal strain and antibiotic consumption in the host (Fig 4A). If a host consumes

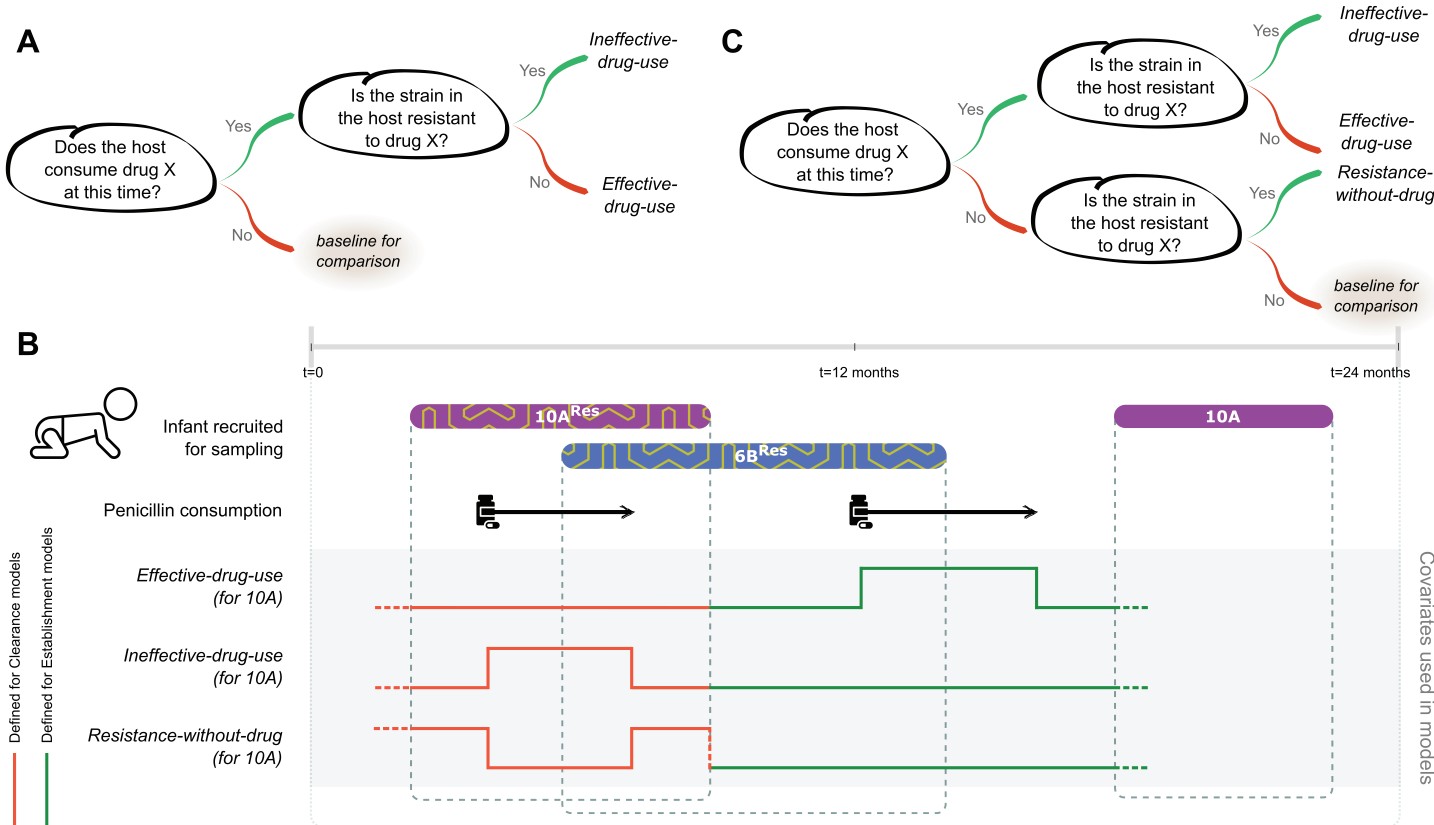

**Fig 4. Definitions of covariates relating to drug usage and antibiotic resistance.** (A,C) Schematics explaining how the categorical variables *effective-drug-use*, *ineffective-drug-use* and *resistance-without-drug* are defined based on the set of drugs consumed by the host and resistances carried on strains. Since these three variables are defined analogous to levels of a categorical variable, their effect sizes are to be interpreted relative to the fitness effect corresponding to '*baseline for comparison*'. (B) An example showing how the values of the three covariates vary with time. The first two episodes of carriage are resistant to penicillin. The orange lines indicate the value of the covariates for the first 10A serotype as used in clearance models. The green lines indicate the value for the second 10A serotype as used in establishment models.

antibiotics that the strain is resistant to, we refer to the drug usage as *ineffective-drug-use*; the variable measures the effect of antibiotics on a resistant strain. If the host consumes antibiotics that the strain is susceptible to, we refer to it as *effective-drug-use*; the variable measures the effect of antibiotics on a susceptible strain. These variables quantify the effects of drug usage on strain fitness relative to the 'baseline' condition (Fig 4A) corresponding to fitness of strains that experience no drug pressure. Both the variables are defined with respect to the focal strain (Fig 4B).

To analyse the effects of antibiotic consumption on clearance rate, we use serotype, *presence-of-competitor*, *effective-drug-use*, and *ineffective-drug-use* as covariates and run a time-varying Cox model on time to clearance data (these covariates produce the lowest AIC compared to a subset of these covariates - See Note I). The hazard coefficient of *effective-drug-use* on clearance is 0.53 (CI = 0.41 to 0.66) (Fig 5A). This shows that consumption of drugs is associated with a 71% (hazard ratio = 1.71, CI = 1.51 to 1.93) increase in the clearance rate of susceptible strains compared to no drug consumption. The hazard coefficient of *ineffective-drug-use* is also positive at 0.34 (CI = 0.14 to 0.53, and hazard ratio of 1.40, CI = 1.15 to 1.70), suggesting that drug consumption increases clearance even for strains resistant to that drug. However, the hazard of *effective-drug-use* on clearance is higher that that of *ineffective-drug-use*, suggesting resistance does reduce drug-induced clearance (the size of this reduction is sensitive to our assumption about the length of all antibiotic treatments being 7 days, and is significant for 10 or more days; see more in Note G in S1 Supplementary information). Overall, the above analyses show that antibiotics clear both susceptible strains and resistant strains, but that resistance is associated with a reduced effect.

The effects of antibiotic use on clearance rate depend on the specific antibiotic consumed. In Note E in S1 Supplementary information, we show the results of the above analyses stratified by antibiotic. For amoxicillin/ampicillin, the hazard coefficients of *effective-drug-use* (0.55, CI = 0.41 to 0.68) and *ineffective-drug-use* (0.34, CI = 0.13 to 0.54) are positive. This is also the case for all other antibiotics, but the confidence intervals overlap zero (Fig Fb in S1 Supplementary information). Amoxicillin and ampicillin, which make up most of the consumption in the dataset (2024 prescriptions during 2441 episodes of treatment), likely drive the overall effect of drugs we see.

## Costs of resistance in the absence of antibiotics

Resistance to antibiotics is potentially associated with a fitness cost, which could affect various components of fitness, including the transmission rate, clearance rate and/or within-host competitiveness. Here, we quantify the effects of resistance on clearance rate and within-host competitiveness in the absence of antibiotics (see Note F in S1 Supplementary information for antibiotic-use-blind analyses of the effects of resistance). We cannot make inferences about transmission rate, as observed establishment rates are dependent on the overall prevalence of resistant and susceptible strains in the population, for which we do not have reliable data.

We expand the categorical variables used in the preceding section to include a new variable for antibiotic resistance of the focal strain (see Fig 4C) called *resistance-without-drug*. Note that the 'baseline' scenario in Fig 4C is different from that used earlier in Fig 4A. *Resistance-without-drug* tracks both the treatment status of the host over time and its relation to the resistance of the focal strain (Fig 4B).

We first ask whether resistance imposes a cost on clearance rate, i.e., whether resistant strains are cleared faster than susceptible strains in the absence of antibiotics. The inference of such a cost, however, is complicated by an epistatic interaction between resistance and duration of carriage, which leads to resistance being more common on long-carried strains [42].

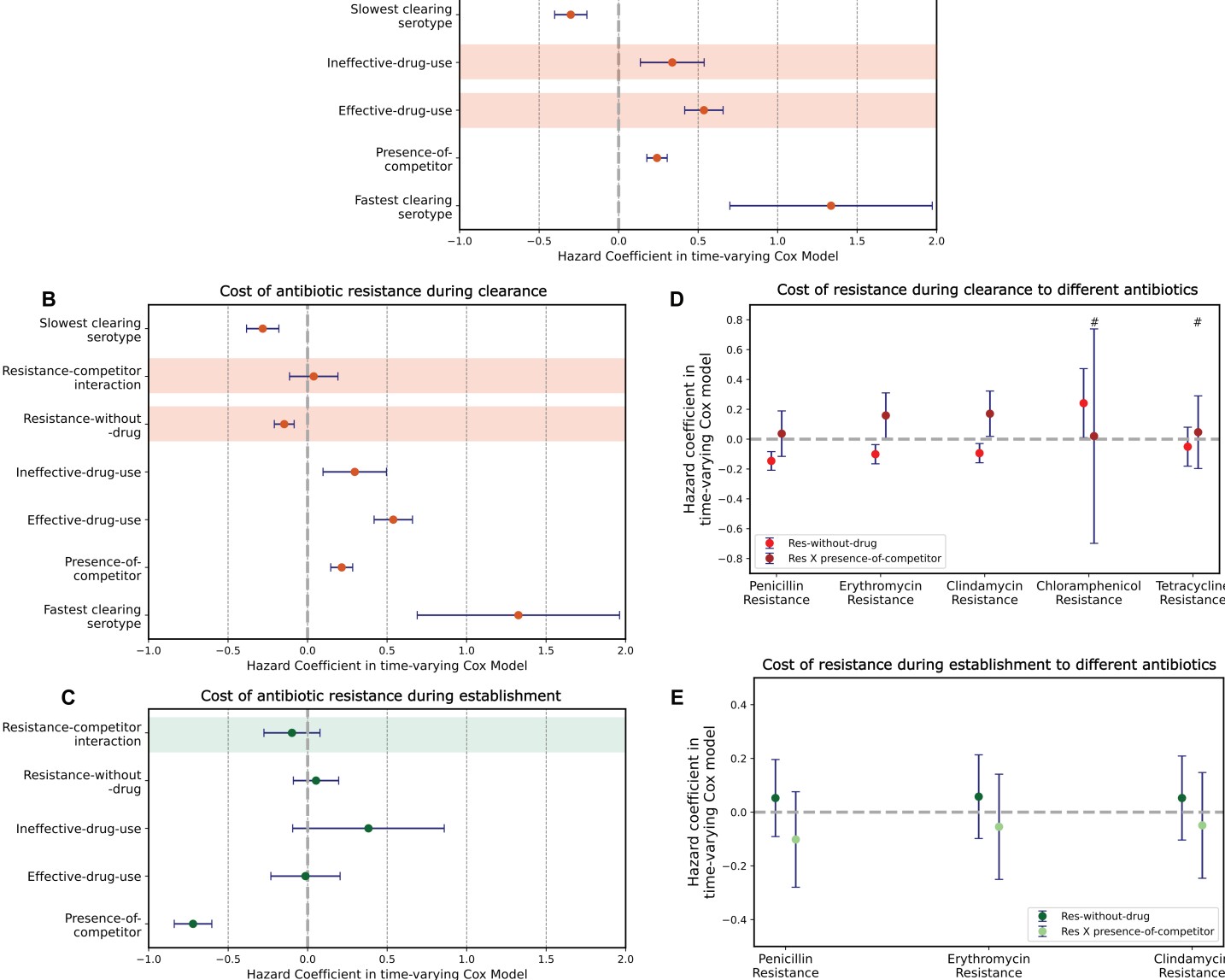

**Fig 5. Models that show the effects of antibiotics and resistance on pneumococcal dynamics.** A,B,D are models for clearance rate, while C,E show models for establishment rate. Error bars show 95% confidence intervals. (A) Does antibiotic consumption affect the clearance rate of resistant and susceptible strains? Here, *effective-drug-use* and *ineffective-drug-use* have positive hazards on clearance, indicating that antibiotics increase the rate of clearance of both susceptible and resistant strains. (B) Does resistance have a cost on clearance or within-host competitiveness? *Resistance-without-drug*, which corresponds to resistance on strains that are not exposed to antibiotics, has a negative clearance hazard, indicating reduced clearance rates. The interaction term between *resistance-without-drug* and *presence-of-competitor* which captures the competitive cost of resistance has a positive non-significant hazard coefficient. (C) Does resistance have a cost on within-host competition during establishment? The interaction term here captures the competitive cost of being resistant rather than susceptible when establishing into an occupied host niche. (D) The analysis in B is repeated for different resistances. # corresponds to effect sizes from models that do not include *effective-drug-use* and *ineffective-drug-use* since these resistances have no corresponding drug treatment in the dataset. See more in Note F in S1 Supplementary information. Only those resistances with enough data for model convergence are shown. (E) The analysis in C is repeated for resistance to different antibiotics. The data underlying this figure can be found in https://doi.org/10.5281/zenodo.15800560.

The ability to detect a cost on clearance rate hinges on the relative strength of this cost compared to the association of resistance with low clearance rates.

To measure the effect of resistance on clearance rates, we use a time-varying Cox proportional hazards model with the following explanatory variables: serotype of focal strain, *presence-of-competitor*, *effective-drug-use*, *ineffective-drug-use* and *resistance-without-drug* (we use this model since these covariates produce a lower AIC than models with a subset of these covariates - see Note I in S1 Supplementary information). The hazard of *resistance-without-drug* here indicates the effect of a resistant strain on clearance rate compared to a susceptible strain in the absence of antibiotics. The hazard coefficient of *resistance-without-drug* is -0.14 (CI = −0.20 to −0.08) (Fig I in S1 Supplementary information, also Fig 5B). This indicates that resistant strains are associated with a 13% lower clearance rate than susceptible ones (hazard ratio of 0.87, CI = 0.82 to 0.92). Thus, any potential cost of resistance here is fully masked by the above-mentioned association. We additionally perform this analysis for each resistance separately to measure costs of resistance to different antibiotics. We find similar negative coefficients for resistance to penicillin, erythromycin and clindamycin (Fig 5D). However, resistance to chloramphenicol is associated with a positive coefficient of 0.24 (CI = 0.01 to 0.47) corresponding to a 27% (hazard ratio of 1.27, CI = 1.01 to 1.60) higher clearance rate of chloramphenicol resistant strains compared to susceptible strains.

Next, we analyse the effect of antibiotic resistance on the competitive ability of co-colonised strains. This competitive cost of resistance can manifest as an increased effect of the presence of a within-host competitor on either the clearance rate or the establishment rate of a resistant strain (compared to a susceptible strain).

To test whether resistance is associated with an increased effect of competitors on clearance of the focal strain, we use a time-varying Cox model with the following covariates: serotype of focal strain, *presence-of-competitor*, *resistance-without-drug*, *ineffective-drug-use*, *effective-drug-use*, and additionally an interaction term between *resistance-without-drug* and *presence-of-competitor*. This interaction term captures how the hazard of clearance of an antibiotic resistant strain in an antibiotic-free host changes due to the presence of a competing strain, thereby reflecting the competitive cost of resistance. The interaction term has a hazard coefficient of 0.04 (CI = -0.11 to 0.19) (Fig 5B): there is an overall 4% (hazard ratio of 1.04, CI = 0.90 to 1.20) change in clearance rate of resistant strains under within-host competition compared to susceptible strains. We also repeat this analysis for resistance to each antibiotic separately where possible (Fig 5D). Here, we find some evidence for a cost of resistance on competitive ability during clearance. Resistance to erythromycin and clindamycin is associated with an increase in the clearance rate of a strain in the presence of competitors of 17% (hazard ratio of 1.17, CI = 1.01 to 1.36) and 19% (hazard ratio of 1.19, CI = 1.02 to 1.38) respectively.

Finally, we test whether resistance is associated with an increased effect of competitors on the establishment of the focal strain, using a time-varying Cox model that includes *presence-of-competitor*, *resistance-without-drug*, *ineffective-drug-use*, *effective-drug-use*, and an interaction term between *resistance-without-drug* and *presence-of-competitor* as covariates. The interaction term here has a hazard coefficient of -0.10 (CI = -0.27 to 0.08) (Fig 5C), corresponding to a 10% (hazard ratio of 1.10, CI = 0.92 to 1.24) decrease in establishment rate of resistant strains into occupied host niches compared to susceptible strains. The analysis for resistance to different antibiotics separately is shown in Fig 5E. Here again, the interaction terms are negative and non-significant for all resistances tested. Overall, we do not see a clear evidence for cost of resistance on competitive ability during establishment.

## Discussion

**Competition affects clearance and establishment.** We analysed the largest-to-date longitudinal dataset on pneumococcal carriage in infants to quantify the effects of the presence of within-host competitors on the clearance rate (or the duration of carriage) and establishment rate of serotypes. The presence of another serotype in the host was associated with a 33% (CI = 24% to 42%) higher rate of clearance and a 54% (CI = 49% to 59%) lower rate of establishment by new serotypes. Our results support the hypothesis that serotypes compete with each other to persist within a host. The competitive effect observed on establishment rate is in line with previous studies which have reported reduction in the rates of co-colonisation of strains compared to rates of initial colonisation [19,43]. These estimates are derived from fitting longitudinal data to mathematical models and range from around 50% to 90% reduction in establishment rates depending on the dataset used, serotypes tested and assumptions made in the model. The competition during establishment is analogous to the term *colonisation resistance* [44,45] mentioned in other contexts, which refers to how resident bacteria reduce the likelihood of colonisation by new bacterial species or strains.

The effect of competition on clearance, on the other hand, has not been extensively discussed in previous work on pneumococcal dynamics. Few studies (with much smaller sample sizes) reported no significant effects of the presence of competitors on the clearance of serotypes [46,47]. Auranen et. al. [47] analysed an epidemiological dataset similar to this study using a multi-state Markov model. They reported a 10-times reduction in establishment rate due to within-host competition, but clearance rate decreased by a factor of 0.81 (CI = 0.48 to 1.26) due to competition. While mathematical models commonly account for a lower rate of co-colonisation using an efficiency factor on transmission rate, they often assume constant clearance rates independent of the presence of co-colonisers [17–20]. Our work emphasizes the significant hazard of within-host competition on accelerating clearance and its potential importance in pneumococcal dynamics.

**Serotype hierarchies during within-host competition.** We find that within-host competitive effects can be predicted by the identity of the competing serotypes. Biochemical properties of the polysaccharide capsule that determine a serotype are known to affect its fitness [48]. Longer-carried serotypes have been previously found to prevent co-colonisation better than short-carried serotypes [43]. Our study finds that a similar correlation also exists for competition at clearance, but in the opposite direction: serotypes with longer durations of carriage are associated with a larger reduction in duration of carriage in the presence of another competing serotype. This is surprising since longer serotypes have thicker capsules that increase other aspects of their fitness [48]. Our result suggests that there may exist different mechanisms of between-serotype competition during establishment and during clearance.

The association between longer inherent duration of carriage and faster clearance by competitors suggests a potential trade-off between fitness in co-colonised hosts and fitness in singly colonised hosts, which could partially explain the diversity of serotypes observed in *S. pneumoniae*. The benefit gained from having an advantage during co-colonisation depends on how often a strain experiences co-colonisation, and thus increases with the prevalence of competing strains. As a result, this type of trade-off gives rise to negative frequency-dependent selection and thus contribute to the maintenance of diversity [49]. Within-host competition has been linked to strain-diversity in other species as well [6,50], through different mechanisms. For example, a recent longitudinal analysis of *E. coli* colonisation [6] also finds that the presence of within-host competitors increases clearance rate, and that this effect is greater when the competitors are from the same phylogroup. Stronger within- than between-phylogroup competition would thus contribute to maintaining phylogroup diversity.

Additionally, we find that within-host competitive outcomes depend on the order of colonisation of serotypes in the host: during co-colonisation, resident strains have a lower risk of clearance than invader strains. A recent experimental study showed such priority effects in competition during establishment in *S. pneumoniae* [51]: when inoculated with multiple serotypes, mice carried higher densities of serotypes that were given an initial head-start in colonisation. This may also translate into an advantage in clearance for the early-coloniser if within-host densities post establishment affect duration of carriage. Our analysis provides direct evidence for such a priority effect on clearance rate.

**Antibiotics affect clearance rates.** We show that consumption of amoxicillin or ampicillin is associated with a 73% (CI = 51% to 97%) increase in the clearance rate of susceptible strains. Interestingly, we find that consumption of these drugs is also associated with increased clearance of resistant strains (by 45%, CI = 15% to 69%). This might be because antibiotics can reduce the replication rate of resistant strains, even if the antibiotic concentration is not as high as the minimum inhibitory concentration (based on which resistance is binarized in our analyses). The slower replication in conjunction with other mechanisms (for instance, host immunity or interspecies competition) could accelerate the clearance of resistant strains. The difference in antibiotic-induced clearance between susceptible and resistant strains (by 28 percentage points for amoxicillin/ampicillin) quantifies the benefit of antibiotic resistance. Although we see that resistant strains are cleared slower than susceptible strains by antibiotics, the evidence for this fitness benefit is not strong due to the large confidence intervals. Infants in our data also consume multiple drugs at the same time (19% of treatment episodes), but our methodology is blind to the differential effects of combinations of drugs on strain dynamics, which been described recently elsewhere [52].

**Fitness costs of resistance.** We estimated fitness costs of antibiotic resistance in the absence of antibiotics on (1) the clearance rate of strains, and (2) their competitive ability during clearance and establishment. Overall, the evidence we find is mixed.

The detection of any clearance-associated fitness costs is confounded by the previously described association between antibiotic resistance and long duration of carriage [42,53]. This association is thought to arise because resistance is more beneficial for longer-carried strains [42,54], rather than due to the effects of resistance on the duration of colonisation. Indeed, our results replicate this finding and show, unlike previous analyses, that the association holds even when focusing on untreated hosts. The exception is chloramphenicol resistance, for which we see an association between resistance and shorter duration of colonisation. This could arise due to a large clearance-associated fitness cost for chloramphenicol. However, as we find this effect for only a single antibiotic, it should be interpreted with caution.

We find some evidence for a cost of resistance on within-host competitiveness. Specifically, we find that the presence of a within-host competitor is associated with a greater effect on the clearance rate of erythromycin (17%, CI = 1% to 36%) and clindamycin (19%, CI = 2% to 38%) resistant strains compared to susceptible strains. For the other resistances, as well as for the overall estimate across all resistances, the direction of the effect is consistent with a fitness cost, but the confidence intervals overlap zero. These results qualitatively align with the estimates of within-host competitive advantage of susceptible strains inferred in a previous study, which also found much higher costs for macrolides than penicillins [16].

The difficulty in identifying and quantifying the fitness cost of resistance is intriguing. Resistance frequencies do not appear to be increasing towards fixation (i.e. 100%) [18]. Indeed, a systematic analysis of resistance trajectories in Europe over two decades shows evidence of resistance frequencies stabilising at intermediate values across a range of bug-drug combinations, and no evidence of a wide-spread increasing trend [55]. Such trends strongly

suggest resistance is associated with a fitness cost; otherwise, we would expect to observe increase towards fixation. It is unclear why this is not apparent in this study. One possible explanation is that the cost of resistance is primarily associated with transmission. To detect effects on transmission rate from this dataset, we would need reliable data on the overall resistance prevalence in the study setting. Another possibility is that because fitness cost is a complex and heterogeneous trait, which depends on both environment and genetic background, its magnitude is very difficult to quantify. This is consistent with evidence from other studies on fitness cost, including experimental studies, often being mixed [56].

**Limitations.** It is helpful to highlight some of the limitations of this work. As with all observational studies, we detect associations, not causation. However, when considered in the context of previous work on bacterial ecology, the effects we find have plausible causal explanations. We use a semi-parametric statistical model (Cox proportional hazards), which makes assumptions about the relationship between covariates (e.g. *presence-of-competitor*) and the outcome variable (e.g. *time to clearance*). Specifically, while the model makes no assumptions about the underlying hazard function—e.g. how the probability of clearance changes with time since establishment, it does assume that covariates affect this baseline hazard multiplicatively, and that the impact of multiple covariates also combines multiplicatively. In principle, the relationship may have a different form - e.g. additive effects. While this seems unlikely to affect the direction of detected effects, it would affect estimated effect sizes.

There are also limitations relating to the completeness of available data. Firstly, serotype detection methods are not perfect, therefore within-host serotype diversity is undersampled. The effects we find are generally robust to undersampling of data (see Methods), but associated with greater uncertainty (Fig D in S1 Supplementary information). The exception is the priority effect we detect: here, undersampling exaggerates the difference between invader and resident competitors. We therefore cannot rule out that the effect we observe is driven by undersampling. The priority effect is also sensitive to the assumption that clearance and establishment events happen halfway between sampled time-points. The effect should therefore be considered with some caution. Secondly, we lacked data on the duration of antibiotic treatment. We assumed 7 days based on WHO guidelines; assuming longer treatments led to estimating a stronger benefit from resistance.

**Conclusion.** Using longitudinal data on *S. pneumoniae* carriage in infants, we analysed the effects of within-host competition and antibiotic resistance on pneumococcal epidemiology. Our findings offer key insights into the dynamics of both pneumococcal colonisation and antibiotic resistance.

## Methods

**The dataset.** The Maela Pneumococcal Collaboration collected nasopharyngeal samples from 952 mother-infant pairs from the Maela refugee camp at the Thai-Myanmar border roughly once a month between 2007 and 2010. Excluding dropouts and stillbirths, samples were collected from 737 infants during their first two years of life. The infants in the dataset represent around 12% of all infants in the camp. The samples were serotyped according to the latex sweep method or WHO pneumococcal carriage detection protocols (234 of the initially recruited infants were serotyped by WHO culture and 721 by latex sweeps). 72 different serotypes were identified in the study across both mothers and infants, 68 of them in infants. Pneumococci lacking cell wall capsules were tagged as Non-typeable (NT). A rarefaction analysis is performed (in Fig Ad in S1 Supplementary information) to show the sampling completeness of serotype diversity in infants. Based on the rarefaction curve, we expect only rare serotypes in the population to be missed in our dataset.

Meta-data included information on episodes of illness, hospital visits and antibiotic consumption by hosts. A total of 2441 episodes of illness were recorded and treatment corresponded to one of the following: amox/ampicillin, ciprofloxacin, gentamicin, erythromycin, cloxacillin, ceftriaxone, clindamycin, cefotaxime, TB(tuberculosis) treatment, cotrimoxazole, cefixime, penicillin, nitrofurantoin. Additionally, a subset of samples were chosen for the detection of phenotypic resistance to seven antibiotics. 33.7% of all samples have been tested for resistance to the following seven antibiotics: ceftriaxone, cotrimoxazole, clindamycin, erythromycin, penicillin, tetracycline, and chloramphenicol. Resistance to at least one antibiotic was detected in 30.4% of all samples.

**Data pre-processing.** We include only data of infants in all our analysis. We assume serotype detection to be fully accurate to determine duration of carriage of serotypes. When a serotype is detected in a sample at time $T_1$ when it is not detected in the preceding sample at time $T_0$, then the serotype is assumed to have established at time $0.5(T_0+T_1)$. If a serotype is already detected in the earliest sample collected from the infant, then the establishment is unobserved (censored). Similarly, a serotype detected continuously until sample $T_2$ but not in the succeeding sample at $T_3$ is assumed to be cleared at time $0.5(T_2+T_3)$ (Sensitivity of results to this assumption is discussed in Note B in S1 Supplementary information). Some monthly swabs were missing in the data, but these do not alter the above estimation of establishment and clearance. The duration between an establishment and clearance is the *Time to Clearance* of a strain. To get the *Time to Establishment*, we consider the preceding time of clearance of that serotype until the time of establishment of the focal strain. This is the time period during which the serotype could potentially establish in the host. The above definitions assume that serotypes do not switch during carriage since this is known to be relatively rare [57].

Co-colonisation is detected when two strains have overlapping episodes of colonisation. How often co-colonisation is detected depends on the ability of sampling methods to capture within-host diversity. Since we expect this diversity to be under-represented in our dataset [14,58], we perform sensitivity analyses to undersampling in Note C in S1 Supplementary information. We simulate undersampling by generating subsets of the data by systematically applying a probability p (1–50%) that each detected serotype could be missed in a sample.

To infer the resistance status of the different episodes of carriage in our data, we assume a strain to be antibiotic-resistant if it was labelled as 'INTERMEDIATE' or 'RESISTANT' in the data, and antibiotic-susceptible if labelled as 'SUSCEPTIBLE'. We also assume that resistance is not gained or lost during an episode of carriage. Thus, an episode of carriage defined by its serotype has a constant resistance status - for instance '6B-resistant' or '6B-susceptible' - when resistance is known. Whenever resistance has been measured multiple times during one episode of carriage and there is a conflict in the resistance status of that strain (in around 1.9% of cases), we take the mode of the measurements to be the correct resistance status. If no unique mode exists, the carriage is assumed to be susceptible. These conflicts could be due to measurement errors or potential gain or loss of resistance during colonisation. Our results are not sensitive to these cases, since removing the episodes of carriage with conflicting resistance does not qualitatively affect our results (see Note H in S1 Supplementary information). We were able to infer the resistance status of 1944 out of the 5529 episodes of colonisation in all infants.

To include both drug use and resistance data in our analysis, we create a map between the two based on antibiotic class. *Drug(in italics)*: resistance status =

*amoxicillin/ampicillin*: penicillin,
*erythromycin*: erythromycin,
*cloxacillin*: penicillin,
*ceftriaxone*: ceftriaxone,
*clindamycin*: clindamycin,
*cefotaxime*: ceftriaxone,
*cotrimoxazole*: cotrimoxazole,
*cefixime*: ceftriaxone,
*penicillin*: penicillin

Resistance profiles against the following drug treatments are not known: ciprofloxacin, gentamicin, TB treatment and nitrofurantoin. An X:Y link in the above map is used to signify that when the host carries Y-resistant strains, then consumption of drug X may not affect those strains. For the four drug treatments not included in the above map, we assume there is no resistance.

**Survival analyses.** We use survival analysis to model our data since some establishment and clearance events of strains are censored. All survival analyses are performed in python using the *lifelines* package [59]. We use the Cox proportional hazards model because: (i) we do not have a priori information about the underlying baseline hazard function to inform its shape, and (ii) it allows us to model competition as a biologically interpretable time-varying factor that increases risk of clearance or establishment, which is not easily done in alternatives like accelerate failure time models. Models of clearance include right censoring when clearance events are unobserved, while models of establishment include left censoring when start of establishment periods are unobserved.

Age of the infant at colonisation is added as a categorical stratification variable in all analyses. We do this since age does not follow the proportional hazards assumption to be included as covariates, i.e., its effect on clearance and establishment vary continuously with time. Age can belong to one of four categories: up to 6 months, 6 to 12 months, 12 to 18 months, and above 18 months.

In the analyses, where required, we add 68 serotypes as 67 dummy variables (and one reference serotype 18C, randomly chosen). Thus all serotype coefficients capture hazards relative to serotype 18C. Non-typeable serotypes (NT) are not included in the analysis. Some serotypes can cause model convergence issues due to low total prevalence in the population (technically known as perfect separation). In these cases, these serotypes are removed from the focal serotypes before the analysis but they are included as potential within-host competitors. In this way, 49 serotypes are included in the models explaining the effect of competitors, and 56 serotypes for models describing overall drug and resistance effects. For models that include any resistance-related covariates, we use only those episodes of colonisation whose resistance information is known to run the model. In all cases where an infant consumes multiple drugs, the model assumes that the drugs do not interact with each other and only affect its corresponding resistances as mentioned in the drug-resistance map above. All variables we include in our models as covariates are binary and unit-less.

We use the Akaike Information criterion (AIC) to determine which model to use for each analysis. Our approach is as follows: to estimate the effect of a given variable X, we use a model that includes X and any covariates that reduce the AIC when added to the model. We calculated the AIC in each of the models used in the main text, and show that models

with only a subset of covariates have a higher AIC than the model used in a given analysis in Note I in S1 Supplementary information. Additionally, models with more variables than necessary also increase the AIC (see Note I), and therefore are less parsimonious. In these calculations, serotype is taken to be one categorical variable, so that AIC for models with subsets of serotype dummy variables are not estimated.

To analyse the effect of serotype identity on clearance rate in the presence of competitors, we used a linear regression approach: we performed Orthogonal distance regression between the hazard coefficients of serotype and those of interaction terms between serotype and *presence-of-competitor*. This method differs from Simple Linear Regression by taking into account uncertainty in both dependent and independent variables (minimises errors in both directions). We are interested in whether the slope of this relationship is significantly different from zero.

The analysis of priority effects during clearance in the presence of competitors can be confounded by the fact that longer durations of carriage are more likely to establish first in the host. To correct for this effect, we generated a null distribution of hazard coefficients of *presence-of-resident-competitor* and *presence-of-invader-competitor* by reshuffling the data 100 times. Each time, the episodes of carriage within each infant are reshuffled randomly within the sampling duration. The hazard coefficients inferred from data are compared to the null distributions. The p-value is the fraction of values in the null distribution that is at least as extreme as the hazard coefficient from data.

## Acknowledgments

We thank Lukas Graz from the Seminar for Statistics (SfS) at the ETH Zurich for statistical advice and Christophe Fraser for discussions.

## Supporting information

**S1 Supplementary information**. Combined supplementary text and figures consisting of Notes A–I and Figs A–K.
(PDF)

## Author contributions

**Conceptualization:** Aswin Krishna, Sonja Lehtinen.

**Data curation:** Stephen Bentley, Paul Turner.

**Formal analysis:** Aswin Krishna.

**Funding acquisition:** Sonja Lehtinen.

**Supervision:** Sonja Lehtinen.

**Writing – original draft:** Aswin Krishna, Sonja Lehtinen.

**Writing – review & editing:** Aswin Krishna, Gerry Tonkin-Hill, Thibaut Morel-Journel, Stephen Bentley, Paul Turner, François Blanquart, Sonja Lehtinen.

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
