## [Editor Report · Decision Letter 0]

18 Mar 2025

Dear Dr Lehtinen, 

Thank you for submitting your manuscript entitled "Quantifying the effects of antibiotic resistance and within-host competition on strain fitness in Streptococcus pneumoniae" for consideration as a Research Article by PLOS Biology.

Your manuscript has now been evaluated by the PLOS Biology editorial staff, as well as by an academic editor with relevant expertise, and I'm writing to let you know that we would like to send your submission out for external peer review.

Once your full submission is complete, your paper will undergo a series of checks in preparation for peer review. After your manuscript has passed the checks it will be sent out for review. To provide the metadata for your submission, please Login to Editorial Manager (https://www.editorialmanager.com/pbiology) within two working days, i.e. by Mar 20 2025 11:59PM.

Kind regards,

Roli Roberts

Roland Roberts, PhD

Senior Editor

PLOS Biology

rroberts@plos.org

---

## [Decision Letter · Decision Letter 1]

30 Apr 2025

Dear Dr Lehtinen,

Thank you for your patience while your manuscript "Quantifying the effects of antibiotic resistance and within-host competition on strain fitness in Streptococcus pneumoniae" went through peer-review at PLOS Biology. Your manuscript has now been evaluated by the PLOS Biology editors, an Academic Editor with relevant expertise, and by two independent reviewers.

You'll see that both reviewers are very positive. Reviewer #1 has some questions about the sampling (which may involve some minor analyses), asks whether sensitivity/resistance could change without a change in strain (again, potentially more analyses), whether there could be within-host serotype switched, and asks you to frame the study more open-endedly. Reviewer #2 asks for some presentational improvements, better consistency of stats reporting, justification of the Cox model, and a “limitations” section.

In light of the reviews, which you will find at the end of this email, we are pleased to offer you the opportunity to address the comments from the reviewers in a revision that we anticipate should not take you very long. We will then assess your revised manuscript and your response to the reviewers' comments with our Academic Editor aiming to avoid further rounds of peer-review, although we might need to consult with the reviewers, depending on the nature of the revisions.

**IMPORTANT - SUBMITTING YOUR REVISION**

*Resubmission Checklist*

*Published Peer Review*

*PLOS Data Policy*

*Blot and Gel Data Policy*

Sincerely,

Roli Roberts

Roland Roberts, PhD

Senior Editor

PLOS Biology

rroberts@plos.org

REVIEWERS' COMMENTS:

Reviewer #1:

In this manuscript, Krishna et al. use a large longitudinal dataset of infants colonized with different strains of S. pneumoniae, with varying levels of antibiotic resistance. Using statistical models, they make inferences about the degree of competition among strains, that is to say how the presence of one strain affects the time to clearance or colonization by other strains, and how such effects vary depending on antibiotic treatments and resistant bacteria. The combination of a unique dataset, thoughtful analysis, and clear writing in the context of the literature make this an important and interesting contribution. The authors clearly show large effects of within-host competition increasing clearance rates and reducing establishment of incoming strains. By contrast, they find smaller effects of antibiotic resistance and relatively small fitness costs of resistance. Counter-intuitively, they find a negative association between a strain's inherent rate of clearance and its change in clearance rate due to competition. This suggests an evolutionary tradeoff between colonization ability and competitive ability - a hypothesis that could be tested with future datasets and experiments.

In general, the analyses are well-explained and the conclusions are well-supported, with appropriate caveats. I am not familiar with Cox hazard models, but these are well-explained and well-justified, and use permutation tests are done to account for possible confounds.

Specific comments:

1) I do have some questions about the underlying data: can we be sure that strain diversity is exhaustively sampled, and does sampling effort vary over time? Are the analyses sensitive to assumptions about sampling completeness, i.e. if diversity is undersampled (or if certain strains are undersampled, etc) would this bias the results in any way? A bit more explanation of the sampling methods, and perhaps something akin to collector's curves (or rarefaction analysis) showing the completeness of sampling would be useful.

2) A related question: it is implied that each strain (e.g. 19F) is matched with a resistance phenotype (e.g. 19F-R and 19F-S) but it is unclear if this is the case, or if resistance phenotypes are not linked to serotype identity. Please clarify.

3) There is an implicit assumption that resistance or sensitivity does not evolve within hosts, but is there justification for this assumption - either from the literature or based on this dataset? In principle, it seems possible that antibiotic treatment could actually select for de novo resistance within hosts, which would be observed as a switch of S to R without a change in serotype. Does this occur? Maybe it is negligible, but it is a hypothesis that could be tested, which could also help explain some of the variation in resistance over time and acros patients.

4) A related question is whether there could be within-host serotype switches, or if this is known from the literature to be negligible.

5) Finally, a semantic distinction that I think is important to clarify the hypothesis being tested: in the title, abstract, and throughout, the authors implicitly assume that strains (serotypes) compete, and seek to quantify the strength of this competition. But the paper could be better framed as asking whether competition occurs, and how to quantify it. I think that a neutral model of strains with equal fitness colonization at random is effectively the null hypothesis being considered, but the title and abstract take it almost for granted the competition does occur. I suggest modifying the title to something more like: "Quantifying the effects of antibiotic resistance and within-host competition in Streptococcus pneumoniae." (cutting the term 'fitness' because it is effectively synonymous with competition in the context of this study).

Reviewer #2:

In this paper, Krishna et al. explore competition dynamics within the host for Streptococcus pneumoniae, using one of the largest infant nasopharyngeal swab datasets available. By applying Cox Proportional Hazards models across multiple independent frameworks, the authors provide a quantitative view of strain clearance and establishment rates. Their results suggest that intra-host competition likely plays a key role in pneumococcal persistence, and they also raise interesting points about the potential roles of priority effects and antibiotic resistance genes in shaping strain fitness before and after antibiotic treatment.

As the authors note in the discussion, studies in E. coli and even earlier S. pneumoniae work (e.g., Lourenço et al., Vaccines 2021; Trzciński et al., mBio 2015) have tackled similar questions. However, what sets this study apart is the strength of the statistical modeling and the ability to make chronological inferences with higher resolution. The study is engaging, methodologically solid, and I believe it makes a valuable contribution that is well-suited for PLOS Biology.

I have only a few minor comments, aimed mainly at sharpening the manuscript further:

Results:

* Please make sure all supplementary figures are clearly and consistently annotated — this will help readers navigate the data more easily.

* Lines 189-195: Please rephrase this section to:

"The exponential of the hazard coefficient yields a hazard ratio of 1.62 (95% CI: 1.45-1.80), indicating that antibiotic consumption increases the hazard of strain clearance by 62%."

I recommend applying this style consistently across the paper. Hazard ratios should be reported directly with their confidence intervals, not as percentage increases, to avoid confusion.

* On p-values: if you present absolute p-values in some places (e.g., lines 334 and 275), please keep that approach consistent throughout the manuscript.

Discussion:

* It would be helpful if the authors briefly explain why they chose the Cox model over other options like Accelerated Failure Time (AFT) models or flexible parametric models. Even a short rationale would add transparency and show that the choice was deliberate, not default.

* I also recommend adding a short "Limitations" section. Some of the findings, especially regarding priority effects, could be quite sensitive to how colonization episodes are defined. Flagging this explicitly would help readers interpret the results with the right level of caution and would strengthen the overall credibility of the paper.

---

## [Editor Report · Decision Letter 2]

20 Jun 2025

Dear Dr Lehtinen,

Thank you for your patience while we considered your revised manuscript "Quantifying the effects of antibiotic resistance and within-host competition on strain fitness in Streptococcus pneumoniae" for publication as a Research Article at PLOS Biology. This revised version of your manuscript has been evaluated by the PLOS Biology editors and the Academic Editor.

Based on our Academic Editor's assessment of your revision, we are likely to accept this manuscript for publication, provided you satisfactorily address the following data and other policy-related requests.

a) Please address my Data Policy requests below; specifically, we need you to supply the numerical values underlying Figs 3ABCD, 5ABCDE, S1ABCD, S2ABC, S3B, S4ABCD, S5, S6ABC, S7ABC, S8ABCD, S9, S10ABCDEF, either as a supplementary data file or as a permanent DOI’d deposition. I note that you already have an associated GitHub deposition, but this seems to only contain the raw data. Please could you complete this deposition with the data and code needed to recreate the Figures? Also, because Github depositions can be readily changed or deleted, please make a permanent DOI’d copy (e.g. in Zenodo) and provide this URL (see below).

b) Please cite the location of the data clearly in all relevant main and supplementary Figure legends, e.g. “The data underlying this Figure can be found in S1 Data” or “The data underlying this Figure can be found in https://zenodo.org/records/XXXXXXXX

c) Please make any custom code available, either as a supplementary file or as part of your data deposition (I understand that this may already be in your Zenodo depostion...?).

We expect to receive your revised manuscript within two weeks. 

*Published Peer Review History*

*Press*

Sincerely,

Roli Roberts

Roland Roberts, PhD

Senior Editor

rroberts@plos.org

PLOS Biology

DATA POLICY:

Regardless of the method selected, please ensure that you provide the individual numerical values that underlie the summary data displayed in the following figure panels as they are essential for readers to assess your analysis and to reproduce it: Figs 3ABCD, 5ABCDE, S1ABCD, S2ABC, S3B, S4ABCD, S5, S6ABC, S7ABC, S8ABCD, S9, S10ABCDEF. NOTE: the numerical data provided should include all replicates AND the way in which the plotted mean and errors were derived (it should not present only the mean/average values).

CODE POLICY

DATA NOT SHOWN?

---

## [Editor Report · Decision Letter 3]

7 Jul 2025

Dear Dr Lehtinen,

Thank you for the submission of your revised Research Article "Quantifying the effects of antibiotic resistance and within-host competition on strain fitness in Streptococcus pneumoniae" for publication in PLOS Biology. On behalf of my colleagues and the Academic Editor, Arjan de Visser, I'm pleased to say that we can in principle accept your manuscript for publication, provided you address any remaining formatting and reporting issues. These will be detailed in an email you should receive within 2-3 business days from our colleagues in the journal operations team; no action is required from you until then. Please note that we will not be able to formally accept your manuscript and schedule it for publication until you have completed any requested changes.

Sincerely, 

Roli Roberts

Senior Editor

PLOS Biology

rroberts@plos.org